# Transforming Neural Network Visual Representations to Predict Human Judgments of Similarity

**Maria Attarian**
Google Research, Brain Team
Mountain View, CA
jmattarian@google.com

**Brett D. Roads**
University College London
London, UK
b.roads@ucl.ac.uk

**Michael C. Mozer**
Google Research, Brain Team
Mountain View, CA
mcmozer@google.com

## Abstract

Deep-learning vision models have shown intriguing similarities and differences with respect to human vision. We investigate how to bring machine visual representations into better alignment with human representations. Human representations are often inferred from behavioral evidence such as the selection of an image most similar to a query image. We find that with appropriate linear transformations of deep embeddings, we can improve prediction of human binary choice on a data set of bird images from 67.8% at baseline to 90.3%. We hypothesized that deep embeddings have redundant, high (4096) dimensional representations; reducing the rank of these representations to 2048 results in no loss of explanatory power. We hypothesized that the dilation transformation of representations explored in past research is too restrictive, and indeed we find that model explanatory power can be significantly improved with a more expressive linear transform. Most surprising and exciting, we find that, consistent with classic psychological literature, human similarity judgments are asymmetric: the similarity of $X$ to $Y$ is not necessarily equal to the similarity of $Y$ to $X$, and allowing models to express this asymmetry improves explanatory power.

Although deep-learning vision models can sometimes predict aspects of human vision [e.g., 4, 5, 17, 6, 8], their behavior often contrasts sharply with human expectations. For instance, small perturbations that are imperceptible to humans can dramatically affect model classification decisions [9]; and texture and local image features drive classifiers [1, 7], whereas humans are more strongly influenced by Gestalt shape. Given that differences exist in how humans and machines represent the world, our goal is to develop techniques that bring their representations into better correspondence. This goal is important for two reasons. First, human vision is robust and visual representations contain a wealth of information about objects and their properties. Bringing representations into alignment might expand the range of tasks for which deep nets are useful [e.g., 12, 18, 26]. Second, if the correspondence is strong, deep nets can serve as a human surrogate for prediction and optimization, allowing us to efficiently determine, say, the best training procedures for people [2, 20, 16, 22].

Let's be more specific about the representations that need to be aligned. In a deep net trained to classify images, the representation in the penultimate layer (prior to the softmax layer) serves as a *deep embedding* of the image. This representation necessarily contains the features essential for discriminating object categories. One might hope to align this representation with the activity pattern in higher cortical areas of the human brain, i.e., a *neural embedding*, but it is not feasible to read out large-scale brain activation at a sufficiently fine spatial and temporal resolution. Instead, one might hope to align *psychological embeddings*—a representation of the features essential for human classification, judgment, decision making, and information processing.

A common method to obtain psychological embeddings requires collecting similarity ratings between pairs of items in a domain and then inferring an embedding in which more similar pairs are closer in the embedding space than less similar pairs. Multidimensional scaling (MDS) [24] has been used

2nd Workshop on Shared Visual Representations in Human and Machine Intelligence (SVRHM), NeurIPS 2020.

for over half a century to obtain psychological embeddings for a fixed set of items whose pairwise similarity matrix is provided. Even at a large scale, it can obtain low-dimensional interpretable embeddings that generalize to behavioral tasks [10]. Although MDS can be used given partial or noisy similarity matrices, it is not productive in the sense that it allows one to predict representations and similarities only for items contained in the original similarity matrix. Ideally, one desires an open-set method, not one that works only for the fixed, previously rated set.

# 1 Background and Related Research

Toward the goal of being able to obtain a psychological embedding for *any* image, methods have been proposed that leverage human similarity judgments in conjunction with deep nets. These nets have the advantage over MDS that they can in principle embed novel images.

Sanders and Nosofsky [22] trained a fully-connected net to re-map from a deep embedding of a pretrained classifier to an MDS representation. They found that the resulting re-mapping generalized well to images held out from the re-mapping training set. While this approach demonstrates that psychological embeddings can be extracted from deep embeddings, the approach is limited in that it produces a representation that is no richer than the MDS embedding. The dimensionality of MDS embeddings is limited by the quantity of human judgment data available; deep embeddings are not.

Peterson, Abbott, and Griffiths [16] bypassed the MDS embedding and used a deep embedding to directly predict human similarity judgments. These judgments, made on image pairs using a 0–10 scale with larger values indicating greater similarity, were placed into a symmetric similarity matrix $\boldsymbol{S}$, where element $s_{ij}$ is the judgment for image pair $i$ and $j$. The matrix is modeled with $\hat{\boldsymbol{S}}$, where

$$\hat{s}_{ij} = \boldsymbol{z}_i^{\mathrm{T}} \, \boldsymbol{W} \, \boldsymbol{z}_j, \tag{1}$$

and $\boldsymbol{z}_i$ and $\boldsymbol{z}_j$ are the deep embeddings for images $i$ and $j$, and $\boldsymbol{W}$ is a diagonal matrix. The parameters of $\boldsymbol{W}$ are obtained by optimization of a squared loss, $||\boldsymbol{S} - \hat{\boldsymbol{S}}||^2$, with an $L_2$ regularizer to prevent overfitting. Peterson et al. found that the best fits to human data are obtained by a VGG architecture [23], whose embedding layer is 4096 dimensional.

Peterson et al. placed no constraint on the sign of the elements of $\boldsymbol{W}$. However, with non-negative elements, $\boldsymbol{W}$ can be decomposed as $\boldsymbol{V}^{\mathrm{T}}\boldsymbol{V}$, and Peterson et al.'s method can be viewed as a form of deep metric learning [11, 13]. That is, the similarity function can be interpreted as computing a dot product of linearly transformed embeddings, i.e., $\hat{s}_{ij} = (\boldsymbol{V}\boldsymbol{z}_i)^{\mathrm{T}}(\boldsymbol{V}\boldsymbol{z}_j)$. In this case, the linear transform rescales individual features (vector elements) of the embedding. Such a transform makes the most sense if these features can be ascribed psychological meaning: when comparing vectors, the rescaling permits some features to matter more, some less. However, the basis used for representing the embedding is completely arbitrary: any rotation of the basis is a functionally equivalent solution to the classifier (because a linear transform of the embedding is performed in the softmax output layer). Consequently, we question whether it is well motivated to restrict transforms to dilating a representation that lies in an arbitrary basis. Peterson et al. likely made this choice because a full $\boldsymbol{W}$ would have 16M parameters and would be underconstrained by the relatively small number of human judgments. We describe a potential solution to this dilemma that both gives the embedding a non-arbitrary basis—hopefully one with psychological validity—and allows us to vary the number of free parameters in the model.

We have two further concerns with Peterson et al.'s method which our solution addresses. First, Peterson et al. $z$-score normalize the embeddings from the pretrained model before using the embeddings to compute similarity. Variance normalization does not matter because any such normalization can be inverted via $\boldsymbol{W}$. However, the zero centering of $z$-scoring alters the angles between embedding vectors, and those angles are not arbitrary: the original classifier uses these angles to compute softmax probabilities. Second, Peterson et al. use absolute similarity ratings for model training. Such ratings are subject to sequential dependencies [15], reducing the signal they convey. Relative judgments—of the form 'is $X$ more similar to $Y$ than to $Z$?'—tend to be more reliable, despite outwardly conveying less information [3, 14, 27].

# 2 Methodology

## 2.1 Data set

We used a previously collected data set of similarity judgments for bird images [20]. The image set contains four bird families (Orioles, Warblers, Sparrows, and Cardinals), four distinct species within

each family, and thirteen distinct images of each species. Examples are presented in Figure 2 of [20]. Mechanical Turk participants were shown a *query* image along with two *reference* images and chose which reference was most similar to the query. See the left edge of Figure 1 for a sample triple. The resulting judgment providess a *triplet inequality constraint* (*TIC*). Some participants were shown the query with *eight* reference images and were asked to choose the *two* most similar. Data from these trials provide 12 TICs: each of the two chosen references is more similar to the query than each of the six non-chosen references. The complete data set consisted of 112,784 TICs.

## 2.2   Models to be evaluated

Figure 1 sketches the structure of our approach. To obtain deep embeddings, we use a headless VGG16 classifier pretrained on ImageNet from the Keras library. The penultimate layer of VGG16 has 4,096 units. To model the TIC, we pass the query and two references through VGG16 and then compute pairwise similarities between the query and each of the references. For a query $q$ and reference $r$, we generalize the learned similarity function of Equation 1 as follows:

$$\hat{s}_{qr} = f(\boldsymbol{z}_q)^{\mathrm{T}} \, \boldsymbol{W} \, f(\boldsymbol{z}_r), \tag{2}$$

where $f : \mathbb{R}^{4096} \to \mathbb{R}^k$ performs dimensionality reduction on the original 4096-dimensional deep embedding. Variants of this model are specified via choice of $f(.)$, $k$, and the constraints placed on $\boldsymbol{W}$. We explore these specific constraints on $\boldsymbol{W}$:

- *Identity*: Use original deep embedding space via $\boldsymbol{W} = \boldsymbol{I}$. This case serves as a baseline.

- *Diagonal*: Rescale the original embedding with a diagonal matrix. In contrast to Peterson et al., We require the diagonal elements to be non-negative by optimizing for an unconstrained diagonal vector $\boldsymbol{v} \in \mathbb{R}^k$ and using $\boldsymbol{W} = \mathrm{diag}(|\boldsymbol{v}|)$. We used this constraint because it seems antithetical to posit that greater alignment of representations should reduce similarity.

- *Symmetric*: Require symmetry of the $k \times k$ matrix, which allows us to interpret the similarity function as applying an arbitrary linear transform to each embedding and then computing their dot-product similarity (see Introduction). We optimize over an unconstrained matrix, $\boldsymbol{V} \in \mathbb{R}^{k \times k}$, where $\boldsymbol{W} = \boldsymbol{V}^{\mathrm{T}}\boldsymbol{V}$. This approach is related to that of Ryali et al. [21], who learn the parameters of a covariance matrix to compute the Mahalanobis distance between representations. We question whether a Mahalanobis distance is the appropriate metric when deep embeddings are trained and used to classify via dot-product softmax functions.

- *Unconstrained*: We optimize directly over an unconstrained $\boldsymbol{W} \in \mathbb{R}^{k \times k}$.

In picking $k$ and $f(.)$, we have two goals. First, we wish to reduce the number of free parameters in our models to avoid overfitting the training data. Second, we wish to impose a basis that is less arbitrary than that of the original deep embedding. As we argued earlier, the basis obtained from training VGG is arbitrary; any rank-preserving linear transform is an equivalent solution given that this transform could be inverted in the softmax layer to achieve the same output. If $\boldsymbol{W}$ is constrained to be diagonal, it seems desirable for the dimensions rescaled by $\boldsymbol{W}$ to have some psychological reality. To achieve these two goals in the simplest manner possible, we treat $f$ as the projection of the deep embedding onto the top $k$ principal components of the embedding space. PCA is performed on

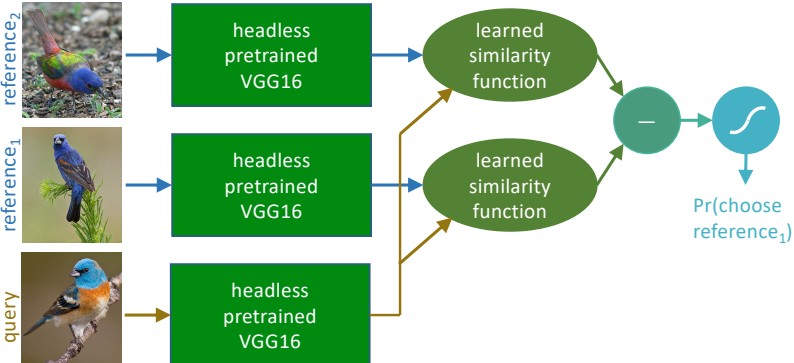

Figure 1: We model human triplet judgments of the form 'Is the query more similar to reference 1 or 2?' After embedding each image, pairwise similarities are computed and the relative similarity determines the probability of choosing one reference or the other.

the embeddings of 23,400 images of the 18 bird classes in the ImageNet training set. Note that these images are distinct from those used for similarity assessment.

To obtain a prediction for the human judgment, we follow a long tradition in choice modeling and assume a logistic function of the relative similarities for query $q$ and references $r_1$ and $r_2$:

$$\Pr(\text{choose ref}_c| \, q, r_1, r_2) = \text{logistic}(\hat{s}_{q,r_c} - \hat{s}_{q,r_{3-c}}). \tag{3}$$

Note that because of the linearity of $f$ and $\hat{s}$,

$$\Pr(\text{choose ref}_c| q, r_1, r_2) = \text{logistic}\left[f(\boldsymbol{z}_q)^{\mathrm{T}}\boldsymbol{W}f(\boldsymbol{z}_{r_c} - \boldsymbol{z}_{r_{3-c}})\right].$$

### 2.3 Simulation details

Models are trained to maximize log likelihood of the training triples,

$$\ell = \textstyle\sum_{(q,r_1,r_2,c)\in\mathcal{T}} \log \Pr(\text{choose ref}_c| \, q, r_1, r_2)$$

where $c$ is the index of the reference chosen by the human rater and $\mathcal{T}$ is the training set. Five-fold cross-validation was performed, yielding 90k and 22.5k triplets in each fold for the training and validation sets, respectively. We trained models using TensorFlow with an SGD optimizer, with Nesterov momentum of 0.9, an learning rate of $10^{-9}$ for the Unconstrained model and $10^{-5}$ for all others. Although trained with likelihood maximization, we evaluate models on *accuracy*, defined as the proportion of examples in which the human response matches the most likely model response. As weak prevention against overfitting, we stopped training when the training accuracy did not improve in the last 10 epochs over the 10 epochs previous.

## 3 Results

Figure 2 shows the outcome of five-fold cross-validation on our data set of human similarity judgments of bird images. The left and right panels show training and validation set accuracy, respectively. A model prediction is scored as correct if the model's probability of selecting the reference chosen by the human is greater than 0.5. Accuracy is plotted as a function of the number of principal-component loadings included in the deep embedding ($k$), and a separate curve is drawn for each different constraint on $\boldsymbol{W}$ that we tested. The dashed blue line is an implementation of Peterson et al.'s method: diagonal weights, allowing negative values, the full 4096-element vector, and imposing an $L_2$ penalty on the weights. To give this model its best shot, we searched for the $L_2$ regularization coefficient over many orders of magnitude that maximized performance on the validation set (i.e., we cheated to benefit this method); we also did not $z$-score the embeddings for reasons explained earlier.

Validation performance roughly tracks training performance. However, for the most complex models (Symmetric and Unconstrained $\boldsymbol{W}$ and large $k$) there appears to be some overfitting: the training curve rises faster than the validation curve. We expected to observe more severe overfitting, manifested by a *drop* in validation performance with $k$, because the largest models have a nearly 185:1 (Unconstrained,

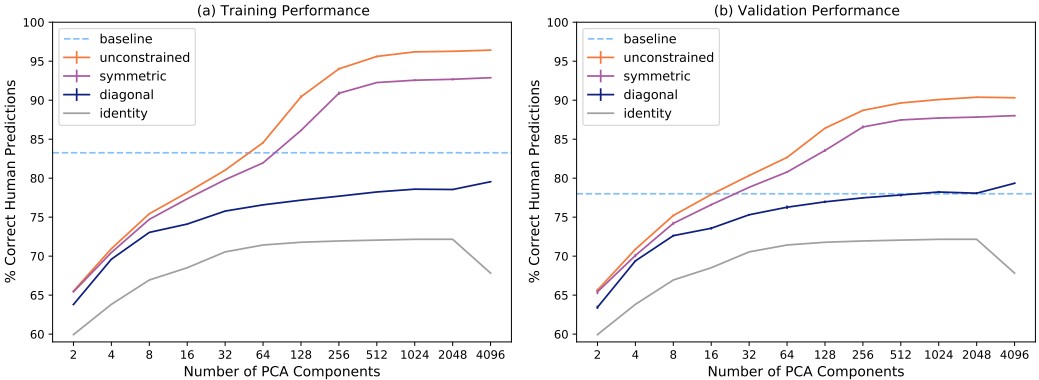

Figure 2: (a) Training and (b) validation performance as a function of the number of principal components included in the embedding ($k$). Each curve represents a different constraint on $\boldsymbol{W}$. Error bars reflect $\pm 1$ SEM on the five-fold cross-validation procedure.

$k = 4096$) and 93:1 (Symmetric, $k = 4096$) ratio of free parameters to training examples (TICs). Contrary to our predictions, the inverted U function is not observed, possibly due to the linear form of the model, which offers a strong constraint on data patterns that can be fit, or to inconsistency in the human judgments that results in violations of transitivity, not fittable by our models. The validation curves in Figure 2b allow us to draw some strong and intriguing conclusions:

- *Dilation of the transformed deep-embedding features (blue curves) achieves a significantly better fit to the human data than by simply using the deep embedding straight from the VGG16 classifier (grey curve).* This result replicates the key finding of Peterson et al. on a new data set and a different response measure. Our work extends Peterson et al. by using the transform $f(.)$ and showing that increasing the dimensionality of the embedding strictly improves the fit to human data, and that adding the non-negativity constraint slightly improves results (for $k = 4096$).

- *Applying a general linear transform to the deep embedding (purple curve) obtains a better fit to the human data than a dilation (blue curves).* Peterson et al. did not investigate using the broader class of transform because it seemed likely that overfitting would occur. However we see a strict improvement in performance with the more complex model, regardless of $k$.

- *Relaxing the symmetry constraint on similarity (i.e., the similarity of A to B does not have to equal the similarity of B to A) improves the fit to human data (red versus purple curve).* This finding was most surprising to us, but in retrospect might have been anticipated by the prominent, longstanding finding that human judgments of similarity cannot be accounted for by the use of an internal psychological distance metric [25]. For example, individuals might judge North Korea to be more similar to China (focusing on the leadership) than China is to North Korea (focusing on the size of the country). To cast this claim in terms of the judgments we are modeling, consider two different triplets: (query $I_1$, references $I_2$ and $I_3$) and (query $I_2$, references $I_1$ and $I_3$). The deep embedding of $I_1$ and $I_2$ are interpreted differently depending on whether they are in the role of query or reference.

In the above results, human-prediction accuracy is assessed on held-out triplets (TICs). To evaluate accuracy for held-out *images*, we ran validation folds in which we randomly selected images to hold out such that the training set was roughly the same size as in our earlier experiment. To best match our earlier experiment, we perform five fold validation (sampling with replacement each fold) for both held-out triplets and held-out images with $k = 2048$. As Figure 3 indicates, models do generalize to new images. The ranking of models is the same, but performance does suffer on new images.

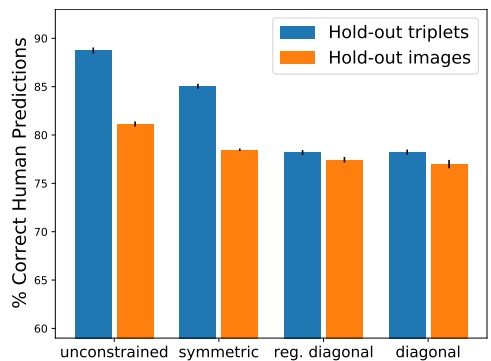

Figure 3: Prediction accuracy for $k = 2048$ with held-out triplets vs. held-out images

## 4 Discussion

Our models of similarity judgment are able to predict 90.3% of human binary choices, suggesting that a deep embedding from a pretrained classifier can be adapted to capture the structure of psychological embeddings of visual images. By applying a linear transform to a deep embedding, we are able to boost the accuracy of prediction from a 67.8% baseline using the original embedding. We significantly improve on an existing approach in the literature [16], which achieves a prediction accuracy on our data set of only 78.0%.

Our simulations reveal several surprising and intriguing results. First, we observe that overfitting is not a serious issue for highly overparameterized linear models—models with forty times as many free parameters as training data points. We are presently investigating whether this finding is due to model linearity (which restricts the transformations that can be applied to the deep embedding) or to some fortuitous early-stopping procedure. Second, and most notably, we observe a benefit for encoding similarity in form that cannot be expressed in terms of distance metric in the embedding space. Rather, one item of a pair is treated as an anchor with respect to which the other item is compared. To the best of our knowledge, researchers in machine learning who model human similar have done so based on distance metrics that do not permit the sort of asymmetry supported by our data [e.g., 12, 19, 27, 26]. Considering the role of anchoring and context in choice is a productive avenue for future research.

## Acknowledgments and Disclosure of Funding

We thank Gamaledin Elsayed for helpful feedback on the manuscript.

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
