# OpenReview forum: "Transforming neural network visual representations to predict human judgments of similarity"
_NeurIPS.cc/2020/Workshop/SVRHM — SVRHM@NeurIPS Poster_

### Official Review · AnonReviewer2 · 2020-10-28
**Review of "Transforming neural network visual representations to predict human judgments of similarity"**

**Rating:** 7
**Confidence:** 4

**Review:**

The authors present a method to transform visual representations from a pre-trained VGG-16 neural network to reproduce human similarity judgments in a binary classification task. This work largely builds on Peterson et al. (2018) in which the same type of transformations were applied to mimic human similarity judgments, but the major contribution is that the authors capture the asymmetry in human judgments similarity judgments, in the classification procedure.

Furthermore, the authors argue that, in Peterson et al. (2018), the embeddings lie in an arbitrary basis and the transformation learned (to match human similarity judgemnts) can be adjusted based on any rotation of the basis. The authors instead choose a "less arbitrary basis" for their embeddings by applying a non-rank preserving transformation to their embeddings -- in this case PCA.


Pros:
- The authors extend the work of Peterson et al. (2018) in an interesting way by allowing the similarity score between a stimulus and referent to be asymmetric, and show this to be the best performing model of human similarity judgments.
- The authors outline how metric learning may not be ideal for modelling human similarity judgments, and provide an alternative way forward.

Cons:
- It's not clear to me why we need the transformation (i.e., the function f) to reduce the rank of the embeddings. The authors mention that if the transformation f is in indeed rank-preserving (as in Peterson et al. (2018)), then the classifier layer of a neural network can invert that transformation, however the authors did not state why that's an issue, especially with regards to modelling human similarity judgments in which the classifier layer is omitted anyways.
- The paragraph on lines 184-194 discussed accuracy for images, but it wasn't clear what was being predicted or how it relates to the human similarity judgments. I'd suggest rephrasing this part since the transition was quite abrupt.


Overall, the paper is well-written and strong quality for a NeurIPS workshop. The authors were mostly clear about the limitations of previous studies and how their work builds on said limitations. This paper followed the lineage of work on modelling human similarity judgments with representations taken from neural network, but showed a new and exciting result. I am not aware of extensive work on modelling asymmetry in human visual representations from a computational perspective, and so the paper is quite novel in that respect.

---

### Official Review · AnonReviewer1 · 2020-10-28
**interesting analysis of including more expressive similarity scores**

**Rating:** 8
**Confidence:** 4

**Review:**

This paper investigates whether neural network activations are limited in their ability to predict human behavior due to the typically imposed linear mapping between the activations and the behavioral similarity metrics. The authors train a similarity mapping directly on human behavioral data using different types of learned similarity functions for the read out layer. The authors surprisingly find that the most expressive models do not overfit, and in fact explain held out human behavioral results the best. The authors also show that their method can incorporate non-symmetrical similarity judgements.

Pros:
* The question of whether transformations should be restricted to linear transforms of the activation space is topical and informative.
* The paper seems like an excellent fit for SVRHM, as it provides insight into how the behavioral similarity metrics between humans and neural networks might be better aligned through more expressive readout layers.
* The presented analysis of different types of constraint matrices was interesting and informative. It is quite interesting that the unconstrained matrix does the best at predicting the similarities in the held out set.

Cons:
* Given the significant drop between the held out triplets and held out images for the larger models, it would be helpful to see the full PCA curve for the held out images, rather than just the result at k=2048 in Figure 3. Is the model overfitting to the specific images that it sees during the training process?
* The paper is motivated by saying that “our goal is to develop techniques that bring [human and machine] representations into better correspondence.” however it is not obvious that this can be achieved by building a different type of read out for the neural network layer (because the underlying representations in the network are still not aligned with human perception). It is unclear whether the authors are arguing that the representation learned via the similarity judgements is the more “human like” part of the network.
* Missing references: Other methods have recently been proposed as ways to make targeted behavioral comparisons between models and humans, specific examples that come to mind include model metamers [Feather et al. 2019] and controversial stimuli [Golan et al. 2019]. It also feels like the paper might benefit from a discussion on triplet networks, contrastive learning and other types of training which directly incorporate similarity judgements during the training process.
* I am skeptical about the optimization process described for training the unconstrained and symmetric models (particularly because of the low learning rate and the nan errors that are documented). It seems like this model still performs the best so perhaps the concern is moot, but the paper would be strengthened by tracking down the optimization problems (for instance, are the gradients exploding, and if so why?)

Questions:
* Do the similarity judgements transfer across different types of images? For instance, is there another dataset (other than birds) on which to evaluate whether the learned similarity metrics transfer? One reason to use only linear transforms on learned feature representations is that these similarity metrics are not dataset specific, and one potential limitation of learning similarity functions directly from human data is that they may overfit  the particular task/data distribution, rather than encompass general ideas about similarity.

---

### Official Review · AnonReviewer3 · 2020-10-29
**Interesting and Relevant Work.**

**Rating:** 7
**Confidence:** 4

**Review:**

This paper investigates how to bring machine visual representations closer to human representations.  This is done by initializing a representation using PCA on features extracted from the penultimate layer of VGG16 and then learning a linear transformation on top to predict triplet choices from human participants. Interestingly, an asymmetric transformation is reported to best capture held out data.

This will be a relevant and interesting contribution to SVRHM.

Comments:

The finding that an asymmetric transformation is most predictive on held out data is interesting and consistent with classical findings. However, it’s unclear how this finding aligns with the aim to bring human and machine representations closer. This is clear for the other transformations considered since they induce a linear transformation of the original representation. In this sense, the asymmetric model seems “overfitted”/ too specific to modeling the triplet task.

While Peterson, Abbott,  Griffiths, 2016, 2018 restricts the learned transformation to a regularized diagonal, other previous work (e.g. Ryali, Wang, Yu, CogSci 2020) has considered more general linear transformations to bring machine representations closer to human representations. On a related note, I’m wondering if you found that the transformed representation better captures hierarchical structure (as was found in Peterson et al)?

The “symmetric” model appears to be more restrictive than symmetric as parameterized - it appears to be positive semi-definite? (of course, this is not to say that it’s not flexible enough to capture arbitrary linear transformation of the original embedding).

It would be nice to have some concrete examples from the data showcasing an example of asymmetry, where the models have divergent predictions.

---

### Decision · Program_Chairs · 2020-11-02

Accept (Poster)